# Design and Compressive Behavior of a Photosensitive Resin-Based 2-D Lattice Structure with Variable Cross-Section Core

**DOI:** 10.3390/polym11010186

**Published:** 2019-01-21

**Authors:** Shuai Li, Jiankun Qin, Bing Wang, Tengteng Zheng, Yingcheng Hu

**Affiliations:** 1Key Laboratory of Bio-based Material Science and Technology of Ministry of Education of China, College of Material Science and Engineering, Northeast Forestry University, Harbin 150040, China; tenacity5856@outlook.com (S.L.); qinjiankun1994@126.com (J.Q.); 18846751347@163.com (T.Z.); 2Science and Technology on Advanced Composites in Special Environments Key Laboratory, Harbin Institute of Technology, Harbin 150001, China; wangbing86@hit.edu.cn

**Keywords:** Variable cross-section core, 2-D lattices structure, analytical model, material utilization, compressive response

## Abstract

This paper designed and manufactured photosensitive resin-based 2-D lattice structures with different types of variable cross-section cores by stereolithography 3D printing technology (SLA 3DP). An analytical model was employed to predict the structural compressive response and failure types. A theoretical calculation was performed to obtain the most efficient material utilization of the 2-D lattice core. A flatwise compressive experiment was performed to verify the theoretical conclusions. A comparison of theoretical and experimental results showed good agreement for structural compressive response. Results from the analytical model and experiments showed that when the 2-D lattice core was designed so that R/r = 1.167 (R and r represent the core radius at the ends and in the middle), the material utilization of the 2-D lattice core improved by 13.227%, 19.068%, and 22.143% when n = 1, n = 2, and n = 3 (n represents the highest power of the core cross-section function).

## 1. Introduction

A lattice structure is a type of neoteric structure composed of two high strength, high modulus thin face sheets, and thick lightweight cores. The reasonable structure, lightweight, and high strength allow the lattice structure to be commonly used in aerospace, shipbuilding, and automobile manufacturing. Improving the structural load capacity, while reducing the weight, has become a main focus within lattice structure research. The main configurations of the lattice structure are the tetrahedron, the pyramid, and the Kagome [1,2,3]. Conventional lattice structure manufacturing methods include the melt foaming method, the powder metallurgical method, the infiltration casting method, and the extrusion-electro discharge method [4,5,6,7,8,9].

There has been substantial research regarding the mechanical properties of lattice structures with different types of cross-sections and configurations. On the mechanical behaviors of lattice structure with uniform cross-section cores, Xiong et al. [10,11] studied the compressive, shear, and flexural behaviors of carbon fiber composite pyramidal truss structures fabricated with the molding hot-press technique (M H-P). Results show that the fabricated low-density truss cores have superior compressive strength. Sun et al. [12,13] studied the compressive and shear behaviors of an improved-pyramidal truss core fabricated with M H-P. Results show that the main compressive failure types are mainly Euler buckling and fracture failure of the struts, while greater deviation exists in shear tests due to the occurrence of dominated node rupture. Wang et al. [14,15,16,17,18,19,20] studied the mechanical behaviors of pyramidal and X-type truss lattice structures. Results show that the mechanical behavior of the pyramidal lattice truss core sandwich face sheets is dependent on the relative density of the core and the material properties of the truss members. The compressive failure types of the X-type are mainly crushed in cores in both ends. Schneider et al. [21] studied a novel manufacturing route to produce fiber composite lattice structures. Results show that single polymer PET cores have better performance compared to high-end foam cores, but have lower performances than carbon fiber lattice cores.

Lattice structures are fabricated by conventional manufacturing methods and great progress has been made in understanding their mechanical behaviors. However, for the lattice structure with variable cross-section cores, the fabrication by conventional methods is not easy and lacks accuracy. Three-dimensional printing (3DP) offers an alternative for making a wide variety of lattice structures with variable cross-section cores, thus making these structures available and accurate for a wide variety of applications. According to the Cannikin Law and the force distribution of the core, a reasonable cross-section design can make additional positions of the core reach load capacity when damaged, which can effectively improve the material utilization of the lattice core. Relatively few studies have investigated the mechanical behaviors of lattice structure with a variable cross-section core. Wu et al. [22,23] studied the mechanical properties and failure mechanisms of polymer sandwich face sheets fabricated by 3DP and the interlocking method with ultra-lightweight 3D hierarchical lattice cores. The study provides insights into the role of structural hierarchy in tuning the mechanical behavior of sandwich structures, and new opportunities for designing ultra-lightweight lattice cores with optimal performance. Rifaie et al. [24] studied the compression behavior of additively manufactured or 3DP polymer lattice structures of various configurations. The lattice structure, based on the body-centered cubic unit cells, was modified by adding vertical struts in different arrangements to create three additional configurations. Results from the experimental data show that selective placement of vertical support struts in the unit cell influence both the absolute and the specific mechanical properties of the lattice structures.

A great deal of work has been performed on the mechanical response of lattice structures with uniform cross-section core fabricated by the conventional manufacturing methods and more accurate 3DP. There is a lack of literature regarding their mechanical behaviors, and the analytical model of lattice structures with a variable cross-section core on analyzing the force distribution of the lattice core. Therefore, designing and manufacturing different types of photosensitive resin-based 2-D lattice structures with variable cross-section cores, and creating a complete set of analytical models represent a worthy research project.

## 2. Design and Fabrication

### 2.1. Core Design

The sizes of the variable cross-section core are shown in Figure 1:

The radius of core y:(1)$$y=\left(R-r\right){\left(\frac{l}{2}\right)}^{-n}{\left|x-\frac{l}{2}\right|}^{n}+r,$$

The moment of inertia of the core:(2)$$I\left(x\right)=\frac{\pi {\left(k{\left|x-\frac{l}{2}\right|}^{n}+r\right)}^{4}}{4},$$

Core cross-sectional area A:(3)$$A=\pi {\left(k{\left|x-\frac{l}{2}\right|}^{n}+r\right)}^{2},$$

Core volume V:(4)$$V=\int_{0}^{l}\pi {\left(k{\left|x-\frac{l}{2}\right|}^{n}+r\right)}^{2}\mathrm{dx},$$
where n is the highest power of core cross-section function.

### 2.2. Unit Cell Design

A 2-D lattice structure with variable cross-section core unit cell schematic illustration is shown in Figure 2. The structural compressive behavior depends on the mechanical behaviors of the raw materials, the length (a), the width (b), and the thickness (t) of the face sheet, the distance (c) between the cores, the length (l), the inclination angle (ω), and the distance (h) between face sheets. The relative density $\bar{\rho }$ of the variable cross-section core is:(5)$$\bar{\rho }=\frac{2\int_{0}^{l}\pi {\left(k{\left|x-\frac{l}{2}\right|}^{n}+r\right)}^{2}\mathrm{dx}}{\mathrm{abh}},$$
(6)a=2(lcosw+c),
(7)h=lsinω,

As seen in Figure 1 and Equation (1), the experiments were divided into three cases to investigate the effects of core cross-section sizes on the structural compressive behavior: n = 1, n = 2, and n = 3, where R = 1.75 mm was a fixed value, r was 1.75, 1.50, 1.25, 1.00 or 0.75 mm, ω=45°, $l=12.5\sqrt{2}\;\mathrm{mm}$, a = 50 mm, b = 25 mm, c = 12.5 mm, and t = 4 mm. The core cross-section functions are shown in Table 1.

### 2.3. Fabrication

2-D lattice structures with different types of variable cross-section cores composed of photosensitive resin (DSM8000, Harbin, China) were designed and manufactured by stereolithography 3D printing (E-Plus-3D S450, Harbin, China), see Figure 3 for the manufacturing processes.

The 2-D lattice structures with variable cross-section cores comprised of eight unit cells were manufactured, and each unit cell had two cores, as shown in Figure 4. The size of the lattice structure was 100 mm × 100 mm × 20.5 mm, including the 16 variable cross-section cores.

## 3. Analytical Model

### 3.1. Force Distribution of Core

As shown in Figure 5a, the structure was ideally simplified into a rod structure with fixed connections between the upper and lower ends. The core in the upper end was subjected to a concentrated load, where it produced a vertical displacement of ∆. The lateral displacement and the rotation angle were both 0. The lateral displacement, the vertical displacement, and the rotation angle of the core in the lower end were 0. A detailed analysis of the uniform cross-section core is found in ref. [25].

The structural mechanics force method was used to solve the force distribution of the variable cross-section core. The processes were as follows:Determine the basic system, as shown in Figure 5b: $X_{1}=M_{A}$, $X_{2}=M_{B}$, $X_{3}=F_{N}$Establish the basic equations of the force method:
(8)$$\delta_{11}X_{1}+\delta_{12}X_{2}+{∆}_{1c}=0,$$
(9)$$\delta_{21}X_{1}+\delta_{22}X_{2}+{∆}_{2c}=0,$$
(10)$$∆\sin\omega =\int_{0}^{l}\frac{X_{3}}{\mathrm{EA}\left(x\right)}\mathrm{dx},$$Determine coefficients and free items:
$$\delta_{11}=\delta_{22}=\int_{0}^{l}\frac{x^{2}}{\mathrm{EI}\left(x\right)l^{2}}\mathrm{dx},\;\delta_{21}=\delta_{12}=-\int_{0}^{l}\frac{x\left(l-x\right)}{\mathrm{EI}\left(x\right)l^{2}}\mathrm{dx},\;{∆}_{1c}={∆}_{2c}=\frac{∆\cos\omega }{l},$$Solve the equation, calculate the rod end bending moment, and the axial force of the core:
$$X_{1}=X_{2}=-\frac{{∆}_{1c}}{\delta_{11}+\delta_{12}},\;X_{3}=\frac{∆\sin\omega }{\int_{0}^{l}\frac{1}{\mathrm{EA}\left(x\right)}\mathrm{dx}}$$

The bending moment was positive in the clockwise direction and the axial force was positive in the compressive stress.

### 3.2. Structural Compressive Behavior

#### 3.2.1. Failure Types

According to Mohr’s stress circle, the axial compression force, the bending moment, and the shear force are coupled, the maximum/minimum axial stress, $\sigma_{1,3}$ is given in ref. [26]:(11)$$\sigma_{1,3}=\sqrt{{\left(\frac{\sigma_{x}-\sigma_{y}}{2}\right)}^{2}+{\left(\tau_{\mathrm{xy}}\right)}^{2}}\pm \frac{\sigma_{x}+\sigma_{y}}{2},$$

There were two structural dangerous locations:The core in middle position had the smallest cross-section area and the axial compressive stress generated by the axial force was the largest. The bending stiffness EI of the core in the middle position was small, resulting in a smaller variable of bending moment, so we approximated that $\tau_{\mathrm{xy}}=0$. With $\sigma_{y}=0$, the maximum axial stress of the core in the middle position was calculated with:(12)$$\sigma_{1}=\sigma_{x}=\frac{X_{3}}{{\pi r}^{2}},$$The bending moment of core at two ends was the maximum, the combined compressive stress generated by the bending moment and the axial force was maximum. The dangerous positions of the core at both ends were at the edge of the cross-section, shown in Figure 2. The area moment of this position was zero, so $\tau_{\mathrm{xy}}=0$. With $\sigma_{y}=0$, the maximum axial stress of the core at the two ends was calculated in ref. [27]:(13)$$\sigma_{1}=\sigma_{x}=\frac{X_{3}}{{\pi R}^{2}}+\frac{4X_{1}}{{\pi R}^{3}{\left(\sin\omega \right)}^{2}},$$
With σ_2_ = 0, the Mises yield stress σ_M_ is given by:(14)$$\sigma_{M}=\sqrt{\frac{{\left(\sigma_{1}\right)}^{2}+{\left(\sigma_{2}\right)}^{2}+{\left(\sigma_{1}-\sigma_{2}\right)}^{2}}{2}}=\sigma_{s},$$
Where σ_s_ is the material yield strength of the 2-D lattice core.

We solved ${∆}_{1}$, ${∆}_{2}$ according to Equations (11)–(14) with the ideal elastic–plastic core and the maximum stress criterion [28].

Displacement ∆ during structure failure was given by:∆ = min (∆_1_, ∆_2_),

When $∆\;={∆}_{1}$, the structure was destroyed by the axial compressive stress in the middle position of the core.

When $∆={∆}_{2}$, the structure was destroyed by the combined compressive and bending load at the two ends of the core.

When ${∆}_{1}={∆}_{2}$, the structure had the most efficient material utilization of the lattice core because the two dangerous positions of the core reached their load capacity simultaneously.
(15)$$\frac{X_{3}}{{\pi R}^{2}}+\frac{4X_{1}}{\pi {\left(\sin\omega \right)}^{2}R^{3}}=\frac{X_{3}}{{\pi r}^{2}},$$

For the slender core, the failure type of the Euler buckling should be taken into consideration.

The discriminant slenderness λ_p_ that was the threshold value of strut slenderness in Euler’s formula was calculated as:(16)$$\lambda_{p}=\sqrt{\frac{{\left(\pi \right)}^{2}E}{\sigma_{p}}},$$
where σ_p_ was the proportional limit of the core material. When the slenderness of the cores exceeded λ_p_, the core was damaged by Euler buckling.

According to the mechanical behavior of the raw material and unit cell sizes, we solved Equations (15) and (16) to obtain the optimized r and λ_p_. The failure mode maps of 2-D lattice structures with variable cross-section cores are shown in Figure 6.

#### 3.2.2. Load Capacity

The structural load capacity depends on the displacement ∆ at the time of structural failure, where the load capacity F_max_ of the 2-D lattice structure with variable cross-section core is given by:(17)$$F_{\max}=N\left(X_{3}\sin\omega +\frac{X_{1}+X_{2}}{l}\cos\omega \right),$$

Note: N is the number of cores, the 2-D lattice structure with variable cross-section core has eight unit cells, each unit cell has two cores, so N = 16.

#### 3.2.3. Equivalent Compressive Elastic Modulus

The equivalent compressive elastic modulus is given by:(18)$$E_{\mathrm{eq}}=\frac{2F\left(h+2t\right)}{\mathrm{Nab}∆}=\left(\frac{{\left(\sin\omega \right)}^{2}}{\int_{0}^{l}\frac{1}{\mathrm{EA}\left(x\right)}\mathrm{dx}}+\frac{2{\left(\cos w\right)}^{2}}{\left(\delta_{11}+\delta_{12}\right)l^{2}}\right)\frac{2\left(h+2t\right)}{\mathrm{ab}},$$

#### 3.2.4. Specific Strength

The specific strength refers to the strength of a unit mass per unit volume, so its value is equal to the ratio of the strength to the density. It is an indicator to measure if the material is light and strong [25]. When the 2-D lattice structure is subjected to flatwise compressive load, the specific strength ${\bar{\sigma }}_{\mathrm{ss}}$ of the 2-D lattice structure with variable cross-section core is given by:(19)$${\bar{\sigma }}_{\mathrm{ss}}=\frac{\left(X_{3}\sin\omega +\frac{X_{1}+X_{2}}{l}\cos\omega \right)∆h}{\rho_{\mathrm{pr}}\int_{0}^{l}\pi {\left(k{\left|x-\frac{l}{2}\right|}^{n}+r\right)}^{2}\mathrm{dx}},$$
where *ρ*_pr_ = 1.16 g∙cm^−3^, is the density of photosensitive resin.

## 4. Experiments

### 4.1. Raw Material Mechanical Properties

Photosensitive resin (DSM8000) was purchased from the Harbin Free Manufacturing Technology Development Co., Ltd., Harbin, China. The compressive, the flexural, and the tensile properties of the photosensitive resin were measured according to the standards of ISO 604-2002, ISO 178-2010, and ASTM D638-2014. The mechanical properties of the photosensitive resin are shown in Table 2.

### 4.2. Flatwise Compression Test

According to the standard ASTM C365-16, the flatwise compressive behavior of the 2-D lattice structures with variable cross-section cores was measured. The experimental values of the structural load capacity were obtained based on the peak of the displacement-force curve. The straight-line segment of the displacement-force curve was used to calculate the experimental values of the equivalent compressive elastic modulus.

## 5. Results and Discussion

### 5.1. Failure Analyses

The three failure types, including 15 types of 2-D lattice structures with variable cross-section cores are shown in Figure 7a. The three varieties of stress–strain curves corresponding to failure types are shown in Figure 7b.

Note: In Figure 7a, the dimensions are denoted in mm, and the core damage position is in the middle. In the purple circle section, the core damage position is located at both ends of the core in the red circle section.

The failure modes of the 2-D lattice structure with variable cross-section cores are shown in Figure 7a, when r was 0.75, 1.00, and 1.25 mm, the failure mode was primarily compressive failure of the cores in the middle position. As r increased to 1.5 mm, the failure modes included the compressive failure in the middle position and the combined compressive bending failure at the two ends. When r = R = 1.75 mm, the failure mode was mostly combined compressive and bending failure at the two ends of the core.

When r = R = 1.75 mm, the core was a uniform cross-section straight rod and the bending stiffness was EI, according to the structural mechanics displacement method: M_A_ = M_B_ = 6EIl^−2^∆cosω, F_N_ = EA∆sinωl^−1^. The bending moment of the core at the two ends was positively correlated to the bending stiffness EI. The bending moment was large at both ends and small near the middle position. According to the symmetry of the core, the middle position of the core was subjected to no bending moment, however, the core was subjected to the same axial force as any of the positions of the cores.

The analysis showed that both ends of the core were subjected to axial force and larger bending moment, but the core in the middle position was only subjected to axial force. The uniform cross-section core was damaged by the combined compressive and bending load at both ends of the core. The core was designed so that both ends were large and the middle was small, which reduced the bending stiffness EI of the core and the bending moment at both ends of the core. The force distribution in the middle position of the core was relatively straightforward and was only subjected to axial force, so reducing the r within a reasonable range could effectively improve the utilization of the lattice core materials. When the r was reduced a small extent, the load capacity of the core at the middle position was greater than at both of the ends. The structure was destroyed by the combined compressive and bending load at the two ends of the core. When r increased to the optimal value (r = 1.5 mm), the two ends and the middle positions of the core reached their load capacity simultaneously. As r decreased to 1.25, 1.00, and 0.75 mm, the load capacity of the core at the middle position was less than at the two ends. The structure was destroyed by the axial compressive stress in the middle position of the core.

### 5.2. Structural Load Capacity, Equivalent Compressive Elastic Modulus, and Specific Strength

The experimental values of flatwise compressive load capacity were obtained based on the peak of the displacement-force curves. The straight-line segments of the displacement-force curves were used to calculate the experimental values of the equivalent compressive elastic modulus. The test results are shown in Table 3 and Table 4.

According to Equation (15), when R = 1.75mm, the optimal r was calculated when n = 1, n = 2, and n = 3, N = 1, r ≈ 1.5 mm, n = 2, r ≈ 1.5 mm, n = 3, r ≈ 1.5 mm.

The theoretical values of the structural load capacity, the equivalent compressive elastic modulus, and the specific strength were obtained from Equations (17)–(19), see Table 5.

A comparison of the theoretical and the experimental load capacity showed good agreement, see Figure 8. When the face sheet size, the core length, and the inclination angle were constant, the cross-section type of core determined the structural load capacity. When R and n were constant, r = 0.75, 1.00, and 1.25 mm, the structural failure mode was mainly compressive failure in the middle position of the cores. The bending stiffness and the compression area of the core in the middle position increased as r increased, resulting in a greater anti-compression capacity of the core in the middle position and an improved structural load capacity. With r continuing to increase, the structural failure modes transformed from compressive failure of the core in the middle position into combined compressive and bending failure at the two ends. This resulted in a higher structural load capacity and a lower rate increase. When R and r were constant, but n was smaller, the lattice structure underwent the same failure type, but the bending stiffness and the volume of core were larger, according to Equation (2). This resulted in a higher structural load capacity.

A comparison of the theoretical and experimental equivalent compressive elastic modulus showed good agreement, see Figure 9. When the face sheet size, the core length, and the inclination angle were constant, the cross-section type of the core determined the structural equivalent compressive elastic modulus. When R and n were constant but r = 0.75, 1.00, or 1.25 mm, the compressive stiffness of the core in the middle position was small and the structure produced a large vertical displacement when it was subjected to a lower flatwise compressive load. This resulted in a smaller equivalent compressive elastic modulus. The compressive stiffness of the core became larger as r increased and the capacity of the structure to resist deformation increased, which increased the equivalent compressive elastic modulus.

According to Figure 10, when the face sheet size, the core length, and the inclination angle were constant, the cross-section type of the core determined the structural specific strength. When R and n were constant but r was smaller, the structure had a lower specific strength. The structural specific strength increased as r increased. When r reached the optimized value (r = 1.5 mm), the structural specific strength reached the maximum value. As r continued to increase, the structural specific strength decreased. When r and R were constant, the structural specific strength increased as n increased.

When r = 0.75 mm, the axial stress of the core in middle position was greater than the combined compressive and bending stress at the two ends of the core. When the middle position of the core was damaged by axial force, the force at both ends of the core was far less than its load capacity, resulting in material waste at both ends of the core and a lower specific strength. When r = 1.00 and 1.25 mm, the combined compressive and bending stress at both ends of the core gradually approached the axial stress in the middle of the core. When the axial stress in the middle of core was equal to the combined compressive and bending stress at both ends of the core (r = 1.50 mm), maximum specific strength and the core utilization were achieved. When r = R = 1.75 mm, the axial stress of the core in the middle position was less than the combined compressive and bending stress at the two ends of the core. When the core at two ends was damaged by the combined compressive and bending stress, the load in the middle position of the core did not reach its load capacity, resulting in material waste in the middle position of core, where the specific strength of the structure and the utilization of the core decreased.

Assuming that the structural load capacity increased by m and the volume of the core material was reduced by n, the degree of improvement in the structural specific strength q is given by:(20)$$q=\frac{1+m}{1-n}-100\%,$$

By comparing Table 5 and Equation (20), when n = 1, n = 2, and n = 3, the structural load capacity of the optimal cross-section core (r = 1.5 mm) was higher than the uniform cross-section core −2.178%, −2.315%, and −2.428%. The volume of core material decreased by 13.605%, 17.959%, and 20.117%, so the specific strength of the structure increased by 13.227%, 19.068%, and 22.143%.

## 6. Conclusions

2-D lattice structures with a variable cross-section core were designed and manufactured via stereolithography 3D printing (SLA 3DP). According to theoretical analysis and experiments, conclusions on the compressive behavior of the 2-D lattice structure with variable cross-section cores were made:Stereolithography 3D printing (SLA 3DP) successfully manufactured a photosensitive resin-based 2-D lattice structure with a variable cross-section core with a high degree of precision and mechanical behavior.When the face sheet size, the core length, and the inclination angle were constant, the cross-section type of core determined the structural compressive response and the failure types.When the 2-D lattice structure with variable cross-section core was subjected to flatwise compressive load, both ends of the core were subjected to large bending moments and axial forces. The middle position of the core was only subjected to axial force. The lattice core was designed so that R⁄r = 1.167, which effectively improved the utilization of the core material. After theoretical analysis, the structural specific strength of the optimal cross-section core was 22.143% higher than its uniform cross-section core.

The analytical model of the 2-D lattice structure with variable cross-section core could be applied to the tetrahedral, the pyramidal lattice, the X-type, and the three-dimensional Kagome lattice structure.

## Figures and Tables

**Figure 1 polymers-11-00186-f001:**
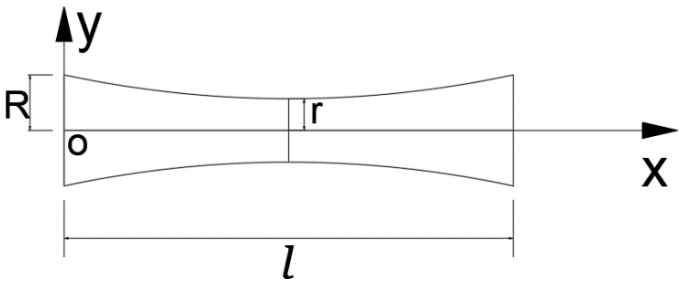
Variable cross-section core schematic illustration.

**Figure 2 polymers-11-00186-f002:**
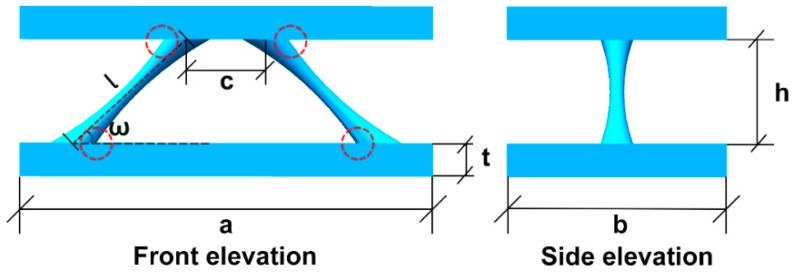
2-D lattice structure with the variable cross-section core unit cell schematic illustration.

**Figure 3 polymers-11-00186-f003:**
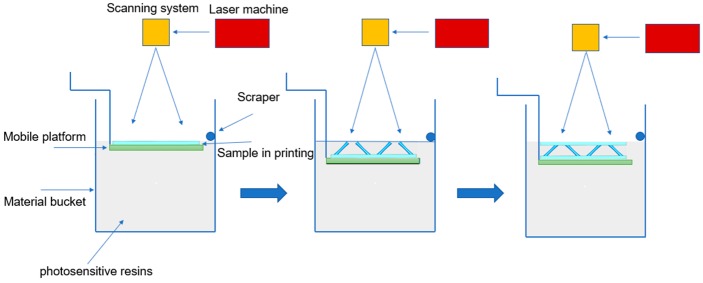
2-D lattice structure with variable cross-section core shaping technology.

**Figure 4 polymers-11-00186-f004:**
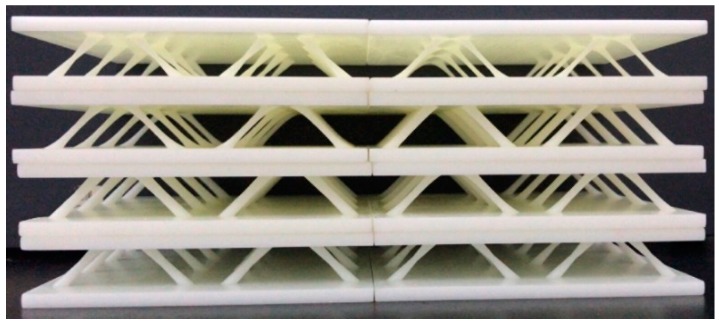
Manufactured 2-D lattice structures with variable cross-section cores.

**Figure 5 polymers-11-00186-f005:**
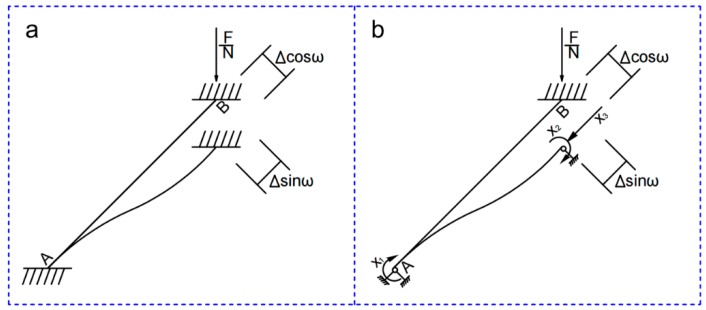
The force method of (**a**) the basic structure and (**b**) the basic system.

**Figure 6 polymers-11-00186-f006:**
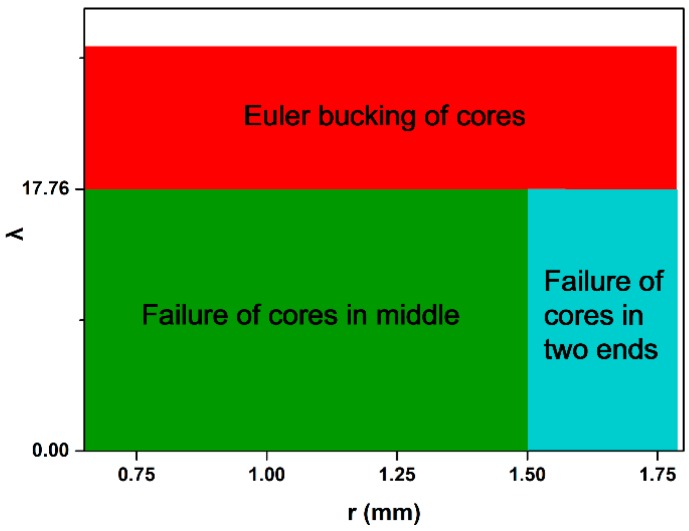
Failure mode maps of the 2-D lattice structures with variable cross-section cores.

**Figure 7 polymers-11-00186-f007:**
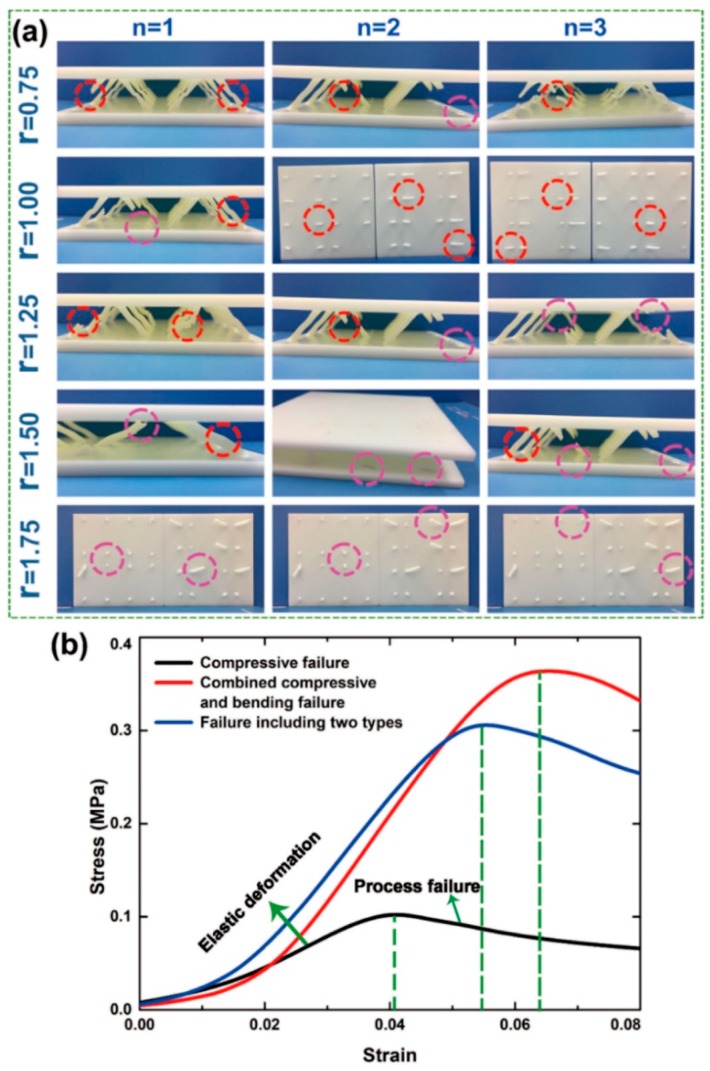
(**a**) 2-D lattice structures with the variable cross-section core failure types and (**b**) compressive stress–strain curves for the 3 failure types.

**Figure 8 polymers-11-00186-f008:**
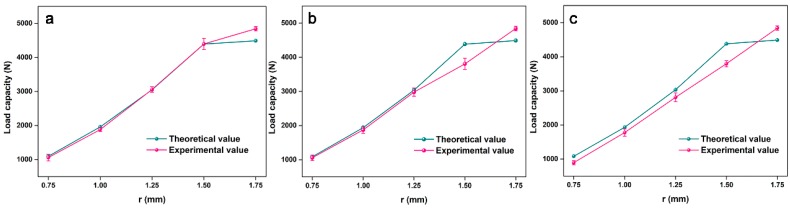
Flatwise compressive load capacity of (**a**) n = 1, (**b**) n = 2, and (**c**) n = 3.

**Figure 9 polymers-11-00186-f009:**
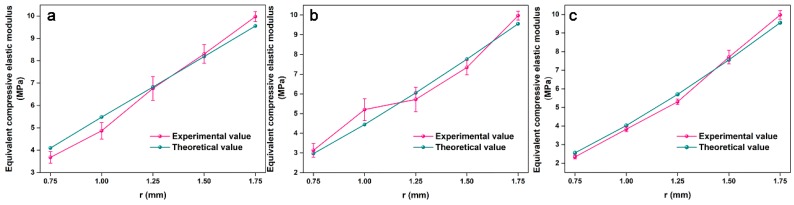
Equivalent compressive elastic modulus of (**a**) n = 1, (**b**) n = 2, and (**c**) n = 3.

**Figure 10 polymers-11-00186-f010:**
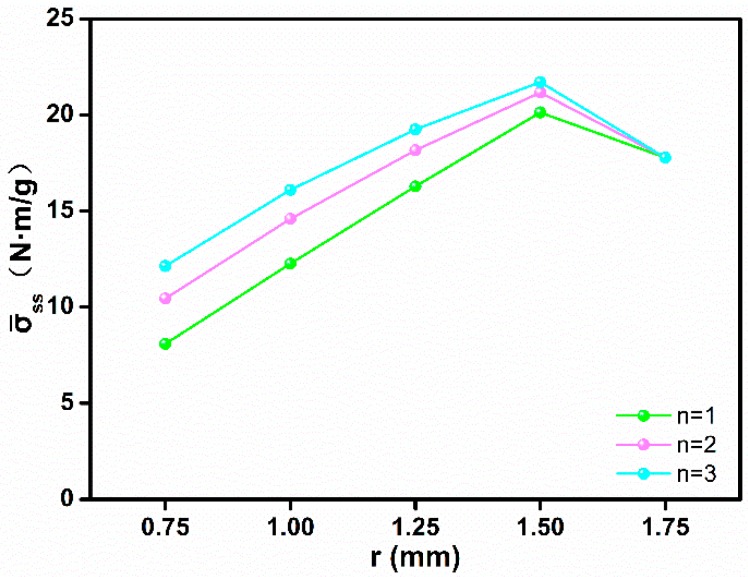
The specific strength of the 2-D lattice structure with the variable cross-section core.

**Table 1 polymers-11-00186-t001:** Core cross-section functions.

n	r (mm)	y (x)
1/2/3	1.75	y(x)=1.75, $\left(0,\frac{25\sqrt{2}}{2}\right)$
1	1.50	$y\left(x\right)=\frac{\sqrt{2}}{50}\left|x-\frac{25\sqrt{2}}{4}\right|+1.50$, $\left(0,\frac{25\sqrt{2}}{2}\right)$
1	1.25	$y\left(x\right)=\frac{\sqrt{2}}{25}\left|x-\frac{25\sqrt{2}}{4}\right|+1.25$, $\left(0,\frac{25\sqrt{2}}{2}\right)$
1	1.00	$y\left(x\right)=\frac{3\sqrt{2}}{50}\left|x-\frac{25\sqrt{2}}{4}\right|+1.00$, $\left(0,\frac{25\sqrt{2}}{2}\right)$
1	0.75	$y\left(x\right)=\frac{2\sqrt{2}}{25}\left|x-\frac{25\sqrt{2}}{4}\right|+0.75$, $\left(0,\frac{25\sqrt{2}}{2}\right)$
2	1.50	$y\left(x\right)=\frac{2}{625}{\left(x-\frac{25\sqrt{2}}{4}\right)}^{2}+1.50$, $\left(0,\frac{25\sqrt{2}}{2}\right)$
2	1.25	$y\left(x\right)=\frac{4}{625}{\left(x-\frac{25\sqrt{2}}{4}\right)}^{2}+1.25$, $\left(0,\frac{25\sqrt{2}}{2}\right)$
2	1.00	$y\left(x\right)=\frac{6}{625}{\left(x-\frac{25\sqrt{2}}{4}\right)}^{2}+1.00$, $\left(0,\frac{25\sqrt{2}}{2}\right)$
2	0.75	$y\left(x\right)=\frac{8}{625}{\left(x-\frac{25\sqrt{2}}{4}\right)}^{2}+0.75$, $\left(0,\frac{25\sqrt{2}}{2}\right)$
3	1.50	$y\left(x\right)=\frac{4\sqrt{2}}{15625}{\left(\left|x-\frac{25\sqrt{2}}{4}\right|\right)}^{3}+1.50$, $\left(0,\frac{25\sqrt{2}}{2}\right)$
3	1.25	$y\left(x\right)=\frac{8\sqrt{2}}{15625}{\left(\left|x-\frac{25\sqrt{2}}{4}\right|\right)}^{3}+1.25$, $\left(0,\frac{25\sqrt{2}}{2}\right)$
3	1.00	$y\left(x\right)=\frac{12\sqrt{2}}{15625}{\left(\left|x-\frac{25\sqrt{2}}{4}\right|\right)}^{3}+1.00$, $\left(0,\frac{25\sqrt{2}}{2}\right)$
3	0.75	$y\left(x\right)=\frac{16\sqrt{2}}{15625}{\left(\left|x-\frac{25\sqrt{2}}{4}\right|\right)}^{3}+0.75$, $\left(0,\frac{25\sqrt{2}}{2}\right)$

**Table 2 polymers-11-00186-t002:** Photosensitive resin mechanical properties.

Compressive Modulus (MPa)	Compressive Strength (MPa)	Flexural Modulus (MPa)	Flexural Strength (MPa)	Tensile Modulus (MPa)	Tensile Strength (MPa)
1699.968	53.170	1897.297	59.285	1205.019	36.460

**Table 3 polymers-11-00186-t003:** Experimental values of the structural load capacity.

r	Experimental Value (N)		
	n = 1	n = 2	n = 3
1.75	4841.900 ± 65.300	4841.900 ± 65.300	4841.900 ± 65.300
1.50	4396.232 ± 164.643	3806.811 ± 164.343	3795.670 ± 89.658
1.25	3057.369 ± 81.650	2978.522 ± 123.720	2809.702 ± 131.432
1.00	1881.722 ± 57.155	1872.668 ± 103.325	1776.109 ± 104.137
0.75	1058.960 ± 97.980	1058.96 ± 81.352	894.255 ± 59.334

**Table 4 polymers-11-00186-t004:** Experimental values of the equivalent compressive elastic modulus.

r	Experimental Value (MPa)		
	n = 1	n = 2	n = 3
1.75	9.976 ± 0.227	9.976 ± 0.227	9.976 ± 0.227
1.50	8.305 ± 0.417	7.343 ± 0.368	7.705 ± 0.368
1.25	6.762 ± 0.534	5.719 ± 0.621	5.303 ± 0.141
1.00	4.868 ± 0.367	5.201 ± 0.557	3.828 ± 0.130
0.75	3.679 ± 0.259	3.133 ± 0.351	2.337 ± 0.098

**Table 5 polymers-11-00186-t005:** The theoretical load capacity, the specific strength, and the equivalent compressive elastic modulus.

n	r (mm)	V (mm^3^)	$\bar{\rho }$ (%)	$X_{1}$ (∆)	$X_{3}$ (∆)	F_max_ (N)	${\bar{\sigma }}_{ss}$ (N·m/g)	$E_{eq}\;(\mathrm{MPa})$
1/2/3	1.75	170.079	2.177	189.742	654.224	4489.428	17.778	9.556
1	1.50	146.939	1.881	162.636	560.764	4391.629	20.129	8.191
1	1.25	126.113	1.614	135.530	467.303	3049.742	16.287	6.826
1	1.00	107.601	1.377	121.114	373.842	1959.093	12.262	5.481
1	0.75	91.403	1.170	81.3180	280.382	1097.907	8.090	4.095
2	1.50	139.534	1.786	147.537	532.124	4385.488	21.167	7.761
2	1.25	112.692	1.442	108.150	415.686	3039.769	18.167	6.052
2	1.00	89.552	1.146	72.532	306.147	1940.480	14.594	4.446
2	0.75	70.114	0.897	42.017	205.265	1087.645	10.448	2.970
3	1.50	135.865	1.739	138.305	138.305	4380.421	21.714	7.554
3	1.25	106.114	1.358	93.255	93.255	3032.192	19.245	5.702
3	1.00	80.825	1.035	55.961	55.961	1932.824	16.106	4.025
3	0.75	60.000	0.768	27.792	27.792	1081.866	12.144	2.554

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
