# Peer review of "Design and Compressive Behavior of a Photosensitive Resin-Based 2-D Lattice Structure with Variable Cross-Section Core"

_polymers, 2019, doi:10.3390/polym11010186_

Reviewer 1 Report

The proposed work treats an interesting topic which is the behavior in out-of plane compression of a lattice  structure where the struts have a variable circular cross-section. 

However, there are several things that are unclear to me and some other that must be improved:

- I could not find within the text a description about what k and n means (starting from equation 1) . What are these? A lot of results that are shown further on are linked to that n but with no explanation.

- Table 1 is showing more than needed. Why should you introduced a column for each omega, l, a, b, c, t, if all of them are kept constant for each of the investigated case?

- Please use face sheets instead of panel when you refer to the core skins (within the introduction, section 2.2 and the rest)

- lightweight is only one word; not light weight

- Section 3.1 has a really strange name... You can just write analytical model.

-  Please don't use the word optimization because you have not done an optimization within your study.

- In equation 17 you note that N is the number of cores. Do you mean the number of unit cells?

- Specific strength should be written instead of specific stress

- English language should be improved. There are long phrases where I lost my self reading them, especially within the introduction section and the results and discussion one.

- Table 3 and table 4 are unclear. What values are you showing in parentheses? 

- You should add a graph that shows how your structure compares with others in terms of stiffness and strength , both related to its density.  For example, see the work of Schneider, C., et al.,  (Compression properties of novel thermoplastic carbon fiber and poly-ethylene terephthalate fibre composite lattice structures. Materials & Design, 2015)

Author Response

Dear reviewer:

Thank you for comments and suggestions. We have completed the review of the manuscript. The revised manuscript has been edited and proofread. We would like to re-submit the revised manuscript to “Polymers” and hope it can be published in this journal.

With kindest regards,

Shuai Li, Jiankun Qin, Bing Wang, Tengteng Zheng, Yingcheng Hu

Detailed response to reviewer:

Question 1: I could not find within the text a description about what k and n means (starting from equation 1). What are these? A lot of results that are shown further on are linked to that n but with no explanation.

Answer: Thanks for your suggestions.

I am sorry for my poor description. In section 2.1, n is the highest time of core cross-section function, k=(R-r) 2n∙l (-n), is the highest term coefficient of the core-section function, which is marked in red font from line 86-87.

Question 2: Table 1 is showing more than needed. Why should you introduce a column for each omega, l, a, b, c, t, if all of them are kept constant for each of the investigated case?

Answer: Thanks for your suggestions.

In section 2.2, constant size is restated, which is marked in red font from line 95-99. The new Table 1 is marked in red font in line 99.

Question 3: Please use face sheets instead of panel when you refer to the core skins (within the introduction, section 2.2 and the rest)

Answer: Thanks for your suggestions.

We have replaced “panel” with “face sheets” in the introduction, section 2.2 and the rest, which is marked in red font in line 28; line 45; line 61; line 91-92; line 246; line 260; line 270; line 306.

Question 4: lightweight is only one word; not light weight

Answer: Thanks for your suggestions.

We have replaced “light weight” with “lightweight”, which is marked in red font in line 28.

Question 5: Section 3.1 has a really strange name. You can just write analytical model.

Answer: Thanks for your suggestions.

We have replaced “Analytical model of structural compressive behavior” with “analytical model”, which is marked in red font in line 111.

Question 6: Please don't use the word optimization because you have not done an optimization within your study.

Answer: Thanks for your suggestions.

We have replaced “optimization” with other proper words.

Question 7: In equation 17, you note that N is the number of cores. Do you mean the number of unit cells?

Answer: Thanks for your suggestions.

    I am sorry for my poor description. N is the number of cores not the number of unit cells, and a unit cell contains two cores, which is marked in red font in line 171.

Question 8: Specific strength should be written instead of specific stress

Answer: Thanks for your suggestions.

We have replaced “specific stress” with “specific strength”, which is marked in red font in line 174-178; line 230; line 243-244; line 270-278; line 282-297; line 312.

Question 9: English language should be improved. There are long phrases where I lost my self-reading them, especially within the introduction section and the results and discussion one.

Answer: Thanks for your suggestions.

   We have made major revision for English language.

Question 10: Table 3 and table 4 are unclear. What values are you showing in parentheses?

Answer: Thanks for your suggestions.

I am sorry for my poor description. In Table 3 and Table 4, for example, “4841.900±65.300”, 4841.900 is the average value, 65.300 is the standard deviation, which is marked in red font in line 235-237.

Question 11: You should add a graph that shows how your structure compares with others in terms of stiffness and strength, both related to its density.  For example, see the work of Schneider, C., et al., (Compression properties of novel thermoplastic carbon fibre and poly-ethylene terephthalate fibre composite lattice structures. Materials & Design, 2015).

Answer: Thanks for your suggestions.

The suggestion of “compares my structure with others in terms of stiffness and strength, both related to its density” benefits me a lot and it is very helpful for the future application of the structure. However, the aim of this paper focuses on studying the effect of the core cross-section size on the structural compressive response and failure modes, then, compare the compressive behavior with uniform cross-section core, finally, provide reference for other scholars’ structural design to improve structural specific stiffness/strength and material utilization. Therefore, we tend to compare my structure with others in terms of stiffness and strength, both related to its density in my following study. I hope you can agree with my opinions, but if you think it is necessary to add the graph, I would do it without hesitation.

We have made major revision for this manuscript. The comments from you help me a lot. We hope the revised manuscript can meet your requirement and can be published in this journal.

Reviewer 2 Report

Dear Authors.

I found your study quite interesting. It presents scientific problem and efficient trial to solve it. It rather concerns mechanical engineering, however, the presented material (cured resin), its properties and description of its behavior in specific mechanical conditions gives permission to publish it in journal concerning polymers. The manuscript is constructed properly – it contain clear and comprehensive introduction, good description in terms of investigated materials and applied methods. Also part concerning analytical simulations is well prepared – assumptions are well described and justified. However, before potential publication you should perform thorougly inspection of the text in terms of language and used terms. Some examples are below:

Abstract: "3x5 kinds"?

Abstract: "the highest material utilization of 2-D lattice core" – probably better would sound "the most efficient material utilization of 2-D lattice core ".

Figure 2: "elevation" – means "view"?

Section 3.1: "Force situation"? You ment "force application", "force distribution"?

Section 3.1: "force method"?

Sincerely

Author Response

Dear review:

Thank you for comments and suggestions. We have completed the review of the manuscript. The revised manuscript has been edited and proofread. We would like to re-submit the revised manuscript to “Polymers” and hope it can be published in this journal.

With kindest regards,

Shuai Li, Jiankun Qin, Bing Wang, Tengteng Zheng, Yingcheng Hu

Detailed response to reviewer:

Question 1: Abstract: "3x5 kinds"?

Answer: Thanks for your suggestions.

We have replaced “3x5 kinds” with “different types” in “Abstract”, which is marked in red font in line 13.

Question 2: Abstract: "the highest material utilization of 2-D lattice core" – probably better would sound "the most efficient material utilization of 2-D lattice core ".

Answer: Thanks for your suggestions.

    We have replaced “the highest material utilization of 2-D lattice core” with “the most efficient material utilization of 2-D lattice core” in “Abstract”, which is marked in red font in line 15-16.

Question 3: Figure 2: "elevation" – means "view"?

Answer: Thanks for your suggestions.

Yes, "elevation" means "view", front elevation has the same meaning as front view.

Question 4: Section 3.1: "Force situation"? You mean "force application", "force distribution"?

Answer: Thanks for your suggestions.

   We have replaced “Force situation” with “force distribution” in section 3.1, which is marked in red font in line 112.

Question 5: Section 3.1: "force method"?

Answer: Thanks for your suggestions.

    Yes, force method (structural mechanics terminology) is a method for solving statically indeterminate structures with unknown generalized forces. Since statically indeterminate structures have redundant constraints, their generalized unknown forces cannot be solved by equilibrium conditions alone.

We have made major revision for this manuscript. The comments from you help me a lot. We hope the revised manuscript can meet your requirement and can be published in this journal.

Round  2

Reviewer 1 Report

All my comments have been addressed.

It remains unclear to me what is with that N (line 171). A unit cell should be the one shown in Figure 2 (which has two struts). Therefore N=16 could be the number of unit cell a sandwich panel has, as you show in Figure 4.

Or, If you refer to Fig. 7, then you should say 16 struts because you have 8 unit cells each one having 2 struts.

It is really difficult to understand the English terms used where you explain the meaning of k (lines 86-87). I suppose there is a mistake when you say times...

Author Response

Thanks for your suggestions.

    I have reinterpreted that N, which is marked in red font in line 170-171. The Eq. (1) was redefined, which is marked in red font in line 28. The meaning of n was reinterpreted, which is marked in red font in line 86.